# Evaluation of Immunogenicity and Clinical Protection of SARS-CoV-2 S1 and N Antigens in Syrian Golden Hamster

**DOI:** 10.3390/vaccines10121996

**Published:** 2022-11-24

**Authors:** Zhenye Niu, Xueqi Li, Yang Gao, Lichun Wang, Shengtao Fan, Xingli Xu, Guorun Jiang, Pingfang Cui, Dandan Li, Yun Liao, Li Yu, Heng Zhao, Ying Zhang, Qihan Li

**Affiliations:** Institute of Medical Biology, Chinese Academy of Medicine Sciences & Peking Union Medical College, Yunnan Key Laboratory of Vaccine Research and Development for Severe Infectious Diseases, Kunming 650118, China

**Keywords:** spike protein, nucleocapsid protein, polypeptide vaccine, Syrian golden hamster

## Abstract

The novel coronavirus (SARS-CoV-2) epidemic continues to be a global public crisis affecting human health. Many research groups are developing different types of vaccines to suppress the spread of SARS-CoV-2, and some vaccines have entered phase III clinical trials and have been rapidly implemented. Whether multiple antigen matches are necessary to induce a better immune response remains unclear. To address this question, this study tested the immunogenicity and protective effects of a SARS-CoV-2 recombinant S and N peptide vaccine in the Syrian golden hamster model. This experiment was based on two immunization methods: intradermal and intramuscular administration. Immunized hamsters were challenged with live SARS-CoV-2 14 days after booster immunization. Clinical symptoms were observed daily, and the antibody titer and viral load in each tissue were detected. The results showed that immunization of golden hamsters with the SARS-CoV-2 structural protein S alone or in combination with the N protein through different routes induced antibody responses, whereas immunization with the N protein alone did not. However, although the immunized hamsters exhibited partial alleviation of clinical symptoms when challenged with the virus, neither vaccine effectively inhibited the proliferation and replication of the challenging virus. In addition, the pathological damage in the immunized hamsters was similar to that in the control hamsters. Interestingly, the neutralizing antibody levels of all groups including immunized and nonimmunized animals increased significantly after viral challenge. In conclusion, the immune response induced by the experimental S and N polypeptide vaccines had no significant ability to prevent viral infection and pathogenicity in golden hamsters.

## 1. Introduction

Since the end of 2019, the novel coronavirus epidemic (coronavirus disease 2019, COVID-19) has swept the world, threatening the lives and health of people of all ages. As of 1 November 2022, over 630 million people have been infected, and 6.5 million people have died [1]. To combat the epidemic, institutions in many countries are developing various types of vaccines, such as inactivated vaccines, protein-based vaccines, adenovirus-vectored vaccines, mRNA vaccines, and attenuated influenza virus vector vaccines [2]. Among these vaccines, the polypeptide vaccine [3] and mRNA vaccine [4], based on different structural forms of the spike protein (S) antigen [3], which is the main epitope of the culprit virus, severe acute respiratory syndrome coronavirus 2 (SARS-CoV-2) [3], have received widespread attention and have elicited significant preventive effects after application [5,6,7]. Comparisons of the immunological characteristics of the population immunized with the polypeptide or mRNA vaccine, the recovered population infected with COVID-19, and the population immunized with the inactivated vaccine have revealed differences in the specific cellular immune responses against the viral core capsid protein (nucleocapsid, N). These differences suggest a need to understand the immunological effects of the viral non-neutralizing antigen structural proteins [8,9,10,11]. In addition, in the COVID-19 patient population, individuals exhibiting strong T-cell responses against the N antigen have the best clinical outcomes [12]. Therefore, it is of clear significance to explore the immunogenicity of effective antigenic components, particularly the N antigens, in COVID-19 vaccines and their interrelationships with S antigens, which induce neutralizing antibodies [13,14]. The use of polypeptide antigen forms to address this question would provide simpler and more definitive results.

This study exploited the advantages of polypeptide antigens by artificially expressing the S1 and N antigens individually or in combination to investigate the ability of the N antigen to induce neutralizing antibodies and influence the response induced by the S1 antigen. The immunological effects of the antigens were analyzed in the golden hamster model susceptible to SARS-CoV-2 infection [15,16], on account of its ACE2 gene sequence being highly homologous with that of human hACE2 as the cell entry receptor of SARS-CoV-2 [17] and showing clinical features, viral kinetics, and histopathological damages similar to those described in many patients with COVID-19 [15,16,17]. Interestingly, our results showed that the presence of the N antigen did not increase the immunogenicity of the S1 antigen, although the N antigen, like the S1 antigen, induced highly titer-specific antibodies that did not exhibit crossover features. More importantly, the presence of these two antibodies (N and S1) alone or in combination did not protect golden hamsters from SARS-CoV-2 infection but alleviated their weight loss after challenge.

## 2. Materials and Methods

### 2.1. Virus and Cells

The novel coronavirus KMS-2 strain (Wuhan) was isolated from COVID-19 patients at the Hospital for Infectious Diseases, in Yunnan Province, in February 2020. Vero cells (ATCC, Manassas, VA, USA) were cultured in high-glucose Dulbecco’s Modified Eagle’s Medium (DMEM; Gibco, NY, USA) containing 10% fetal bovine serum (FBS; HyClone, Logan, UT, USA).

### 2.2. Preparation of the Experimental Protein Vaccine

The proteins (10 μg/dose; >90% purity, host: HEK293; Sanyou Biomedical Co., Ltd., Shanghai, China) were prepared singly (S or N) or mixed (S + N; 5 μg + 5 μg per dose). The protein preparations were mixed with alum adjuvant (0.35 mg/dose) and adsorbed overnight at 4 °C to prepare experimental vaccines.

### 2.3. Animals and Ethics

Twenty-one-week-old Syrian golden hamsters were purchased from Beijing Weitong Lihua Laboratory Animal Technology Co., Ltd., Beijing, China. The experimental animal protocol was reviewed and approved by the Animal Ethics Committee of the Institute of Medical Biology, Chinese Academy of Medical Sciences (approval no.: DWSP202102 051).

### 2.4. Immunization and Challenge Program

Seventy hamsters were randomly divided into seven groups (ten in each group, half male and half female) and immunized with two doses of the S, N, or S + N experimental vaccine at an interval of 14 days via intramuscular (IM) or intradermal (ID) injection (Figure 1). At the same time, a control group immunized with PBS via ID injection was established. At 14 days, after booster immunization (two-dose injection), blood samples were obtained, and the hamsters were challenged with KMS-2 (10^4^ CCID_50_/hamster) via nasal spray in a BSL-3 laboratory. After challenge, clinical symptoms were observed, weights were measured, and nasal and oropharyngeal swabs were obtained daily. On the 4th and 10th days, three animals in each group were euthanized (Figure 1). Blood, bronchoalveolar lavage fluid (BALF), nasal lavage fluid (NLF), and various tissues and organs were obtained from the hamsters.

### 2.5. Neutralizing Antibody Detection

The neutralizing antibody (NAb) titer was detected by microtitration. First, the serum was inactivated at 56 °C for 30 min, followed by twofold serial dilution. Fifty microliters of diluted serum were mixed with live virus (100 CCID_50_/50 μL) and incubated at 37 °C for 2 h. Then, a Vero cell suspension (10^4^/100 μL) was added to the mixture and incubated at 37 °C with 5% CO_2_ for 7 days to observe the cytopathic effects (CPEs). The geometric mean titers (GMTs) of NAb were measured. A titer of 4 was defined as the positive threshold (greater than or equal to 4 indicated positive, and less than 4 indicated negative).

### 2.6. Enzyme-Linked Immunosorbent Assay (ELISA)

An ELISA plate (Corning Inc., Corning, NY, USA) was precoated with SARS-CoV-2 spike receptor-binding domain (RBD) protein expressed and purified from eukaryotic cells (5 μg/well, PNA003; Sanyou Biomedical Co., Ltd., Shanghai, China), SARS-CoV-2 nucleocapsid protein (5 μg/well, PNA006; Sanyou Biomedical Co., Ltd., Shanghai, China), and purified virus total antigen (5 μg/well, prepared by the Institute of Medical Biology, Chinese Academy of Medical Sciences, China) at 4 °C overnight. The plates were blocked with 5% bovine serum albumin (BSA, Sigma–Aldrich, St. Louis, MO, USA). The detected serum was serially diluted twofold, added to the plates (100 μL per well), and incubated at 37 °C for 1 h. After washing five times, an enzyme-conjugated secondary antibody (100 μL per well; SinoBiological (SB), Beijing, China) was added and incubated at 37 °C for 1 h. After washing, 100 μL of substrate chromogenic solution (Solarbio, Beijing, China) was added, and the plate was incubated at 37 °C for 15 min. Finally, 50 μL of stop solution was added. The optical density (OD) value of each well at 450 nm was recorded using an ELISA reader (Gene Company, Beijing, China). Samples with an OD value at least 2.1-fold higher than the mean OD value of the negative control serum were defined as positive. The endpoint titer was defined as the highest dilution of a positive serum sample. The GMT was calculated based on positive serum samples from the same group.

### 2.7. Viral Load Detection

Total RNA from the samples was extracted by TRIzol reagent (Tiangen Biotech, Beijing, China). Real-time RT-PCR assays were performed using a One-StepPrimeScript RT-PCR Kit (Takara, Shuzo, Japan) with a Bio-Rad Real-Time PCR Detection System (CFX96; Bio-Rad, USA). The specific primers for qRT-PCR detection were designed based on the ORF1ab sequence of SARS-CoV-2: F: 5′-CCCTGTGGGTTTTACACTTA-3′, R: 5′-ACGATTGTGCATCAGCTG-3′ and probe: 5′-FAM-CCGTCTGCGGTATGTGGAAAGGTTATGG-TAMRA-3′ [18]. The viral gene copy number in the samples was detected by one-step RT-PCR. The PCR conditions were as follows: reverse transcription at 42 °C for 5 min and 95 °C for 10 s and 40 cycles of 95 °C for 5 s, followed by 60 °C for 30 s. The viral RNA was quantified by reference to a standard curve generated using 10-fold dilutions of RNA standard.

### 2.8. Histopathological Examination

Tissue samples were fixed with 10% paraformaldehyde, embedded in paraffin, and cut into 4-μm-thick sections. The sections were stained with hematoxylin–eosin (H&E) to assess morphology.

### 2.9. Statistical Analysis

GraphPad software was used for statistical analyses. Two-way ANOVA was used to compare differences between groups. A value of *p* < 0.05 was considered significant.

## 3. Results

### 3.1. SARS-CoV-2 Structural Proteins S and N Induce Antibody Responses in Syrian Golden Hamsters

Syrian golden hamsters were immunized with the experimental protein vaccines through two immunization routes: ID and IM. The titers of neutralizing antibodies and IgG antibodies against S, N, and whole virus antigens were detected 14 days after booster immunization. In contrast to immunization with the N protein alone, immunization with the S protein alone or S + N induced NAb production with a GMT of 17.15–51.98 (Figure 2). However, immunization with S + N did not enhance the NAb response compared with immunization with the S protein alone, at 14 days, after both primary and booster immunization (*p* > 0.05). As expected, immunization with the S protein alone did not induce the production of IgG antibodies against the N antigen, and immunization with the N protein alone did not induce the production of anti-S IgG antibodies. Almost all immune groups produced anti-whole-virus antigen antibodies after the first immunization, with a GMT of 131–293.95. Moreover, higher levels of antibody production were induced after booster administration, with GMTs of up to 2785.76 (Figure 2). Similar to the Nab response, there was no significant difference in the antibody titers induced by immunization with the mixed protein (S + N) compared with one protein alone (S or N) via the same immunization route. In addition, one-dose ID immunization induced NAb production slightly better than IM immunization (*p* > 0.99) and two-dose immunization (*p* > 0.41). The induction of anti-S antigen and anti-whole-virus antigen IgG antibody production did not differ significantly between one-dose immunization via ID and IM (*p* > 1). However, two-dose ID immunization with the mixed protein (S + N) was slightly more effective than IM immunization (*p* < 0.05) but not immunization with one protein alone (*p* > 0.38). In terms of anti-N antigen IgG antibody levels, there was no significant difference between ID and IM immunization (*p* > 0.14) (Figure 2).

### 3.2. Immunization Partially Alleviates the Clinical Symptoms of Hamsters in Response to Viral Challenge

Fourteen days after booster immunization, a viral challenge experiment was performed in a BSL-3 laboratory to evaluate the protective effect of the experimental vaccine. A previous study reported significant weight loss among hamsters but no deaths after challenge [15,19]. Thus, the weight of the hamsters was assessed every day as a major protective evaluation index. From 2 to 6 days after challenge, the weight loss of the immunized hamsters was slightly less than that of the control hamsters, although all animals lost weight regardless of vaccine type (S, N, or S + N) and route (ID or IM) (Figure 3). In almost all groups, the weight dropped to the lowest point on the 6th day after challenge and then rebounded significantly on the 7th day. For the ID route, on the 6th day after challenge, the weight change of the immunized hamsters ranged from negative to positive as follows: S protein group (5.5% reduction) > S + N protein group (0.9% reduction) > N protein group (2.4% increase). However, for the IM route, the weight change of the immunized hamsters ranged from negative to positive as follows: N protein group (0.1% reduction) > S + N protein group (4.5% increase) > S protein group (4.9% increase) (Figure 3).

### 3.3. Immunization with S and N Proteins Does Effectively Inhibit Viral Proliferation in Hamsters

#### 3.3.1. Viral Detoxification in Immunized Hamsters during Challenge Is Similar to That of Control Hamsters

After viral challenge, weight loss and the trend of detoxification were similar in immunized hamsters and the control group. From 1 to 4 days after challenge, in the nasal swabs, the viral loads gradually increased to 10^10^ copies/100 μL in all animals from the immunization and control groups, and the viral loads gradually decreased to 10^6^ copies/100 μL from 5 days after challenge (Figure 4). The viral loads in the oropharyngeal swabs of each group of animals were similar over the whole challenge period and were maintained at more than 10^6^ copies/100 μL. In addition, the viral load in the blood of immunized hamsters was similar to that in the blood of the control hamsters, regardless of whether the immunogen was the S protein, N protein, or S/N mixed protein (Figure 4).

#### 3.3.2. Immunized Hamsters and Controls Have Similar Viral Loads in Tissues and Organs during Challenge

To further evaluate the protective efficacy of the experimental vaccines, the viral loads in tissues and organs were detected in all groups. Although the viral loads in the individual tissues of the immunized hamsters were different from those in the control hamsters (*p* > 0.05) at 10 days after infection, the total viral loads in the tissues and organs of the immunized hamsters were similar to those in the control hamsters (Figure 5). In the heart, the viral loads in the control group were higher (10^14^ copies/100 μL) than those in the immunized group (less than 10^8^ copies/100 μL) for both ID and IM administration 10 days after challenge. On the other hand, in the genitals, the viral loads in the control group were lower (less than 50 copies/100 μL, negative) than those in the immunized group (greater than 10^4^ copies/100 μL, positive) for both ID and IM administration.

### 3.4. Pathological Damage in Immunized Hamsters upon Viral Challenge Is Similar to That in Control Hamsters

#### 3.4.1. Pathological Changes in the Lungs

The assessment of the degree of pathological damage to the lung, as a major infected and pathologically damaged tissue, is a key index of the protective effect of experimental protein vaccines. Consistent with the viral loads in the lung, pathological damage was similar in all groups, including the control group (Figure 6 and Figure 7).

#### 3.4.2. Pathological Changes in Other Tissues

The pathological damage to various tissues and organs of immunized hamsters was observed to evaluate the protective effect of the experimental protein vaccines. After challenge, the pathological damage to tissues and organs was similar in all groups (Figure 7). Consistent with reports on SARS-CoV-2 infection, lesions were more prominent in the lungs than in the other organs.

### 3.5. Viral Challenge Significantly Increases Neutralizing Antibody Levels

The Nab response induced in the immunized and control hamsters after challenge was detected. In all groups, the GMT of neutralizing antibodies rose from the original levels (before challenge) to approximately 512 on the 10^th^ day after infection (Figure 8).

## 4. Discussion

According to some reports, the golden hamster is a susceptible model to SARS-CoV-2 and exhibits some clinical signs similar to humans, such as alveolar damage, high viral load in the lungs, and serum antibody formation [15,16,17,20], so it would be a good model to study infection, vaccines, therapeutics, and transmission for SARS-CoV-2. Therefore, this study used this model to evaluate the immune response induced by experimental proteins and the protective effect of immunization with the S1 and/or N antigen alone or in combination via different routes. Immunization with the S protein alone or S + N twice at an interval of 14 days by either the ID or IM route induced neutralizing antibodies against SARS-CoV-2, with GMTs of 17.15–51.98. Significantly, the N antigen did not affect the ability of the S1 antigen to induce neutralizing antibodies, as the levels of the neutralizing antibodies induced by the S1 antigen in the presence of the N antigen were not significantly different from those induced by the S1 antigen alone. This result was further supported by the trend in the levels of IgG antibodies against S or against whole virus in immunized animals.

Given the above data and previous reports that the S1 antigen can induce neutralizing antibody responses [21] and that the N antigen can induce cellular immune responses [18,22], the logical corollary is that the immune response induced by both the S1 and N antigens should be effective against viral infection. However, to our surprise, this immune response failed to provide effective protection when the immunized animals were challenged with the live virus. All clinical symptoms, including weight loss and detoxification in the oropharynx and nose and viremia, were observed in both immunized animals and nonimmunized animals. Additionally, virus replication and obvious pathological damage were detected in important tissues and organs, and the levels of these indicators were nearly identical in immunized and nonimmunized animals. Interestingly, dramatic weight loss was observed in the nonimmunized group but not in immunized animals after challenge (~10% of body-weight loss). These results suggested that the clinical symptoms of infection were partially relieved, even though the immune response induced by the proteins did not adequately suppress viral infection in Syrian golden hamsters.

Our findings differ from the previously reported effects of a SARS-CoV-2 inactivated vaccine in rhesus macaques [23] and golden hamsters. Immunization with the inactivated vaccine also produced the same level of neutralizing antibodies as in this study, with a GMT of approximately 40 (Appendix A) [23]. However, the induced immune response was effective against the virus because there was no significant serious damage either clinically or pathologically (Appendix A). The levels of neutralizing antibodies in the animals after challenge were also approximately 512 (Appendix A). As the levels of the neutralizing antibodies induced by the inactivated vaccine and protein components and the trend of antibody changes after challenge were similar, what caused the different results? The greatest difference is that this study used only two components, namely the S1 and N proteins, while the SARS-CoV-2 inactivated vaccine contains almost all the viral proteins, including neutralizing antigenic proteins, non-neutralizing antigenic proteins, and other proteins [24]. Therefore, it is likely that the different proteins in the inactivated vaccine cooperate with each other or act as natural adjuvants in space and time to stimulate the induction of multidimensional and multifaceted systemic immune responses. In addition, the natural proteins often have multimeric structures that further enhance the immune response. Unfortunately, there are limited reports on the roles of each viral protein [18,22,25,26], especially on the interactions between them and their structures, so further studies are needed.

The results from animal models should be viewed with caution. There are also some reports on the use of golden hamsters to evaluate the protective efficacy of the recombinant adenovirus vector vaccine and protein subunit vaccine for COVID-19 [27,28]. However, even for the protein subunit vaccine that is most similar to the proteins used in this study, the levels of neutralizing antibodies differ due to the different protein regions used for immune induction and the different immune doses, among other reasons. As a result, the final protective effects differ, although the trend of weight loss is similar [28]. The immune response is an extremely complex and exquisite system in which there are not only humoral immune-related indicators but also a series of other indicators, including cytokines, chemokines, and cell populations. Regrettably, commercial reagents for detection in Syrian golden hamsters are limited, so it is difficult to test for the major immunological indicators, which hinders a comprehensive analysis and evaluation of the characteristics of experimental vaccines.

In conclusion, this study observed an interesting phenomenon in which the coexistence of the non-neutralizing antigen (N protein) and neutralizing antigen (S1 protein) failed to prevent viral infection in immunized golden hamsters and only relieved weight loss.

## Figures and Tables

**Figure 1 vaccines-10-01996-f001:**
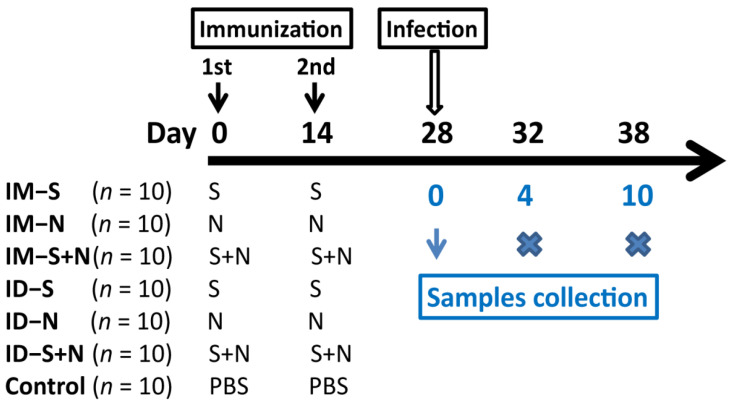
Design of animal experiment. The hamsters were immunized with two doses of the S (IM-S and ID-S), N (IM-N and ID-N), or S + N (IM-S + N and ID-S + N) experimental vaccine at an interval of 14 days via intramuscular (IM) or intradermal (ID) injection. In the control group, the hamsters were immunized with PBS via ID injection. Fourteen days after booster immunization, the hamsters were challenged with KMS-2 in a BSL-3 laboratory. On the 4th and 10th days after challenge, three animals in each group were euthanized.

**Figure 2 vaccines-10-01996-f002:**
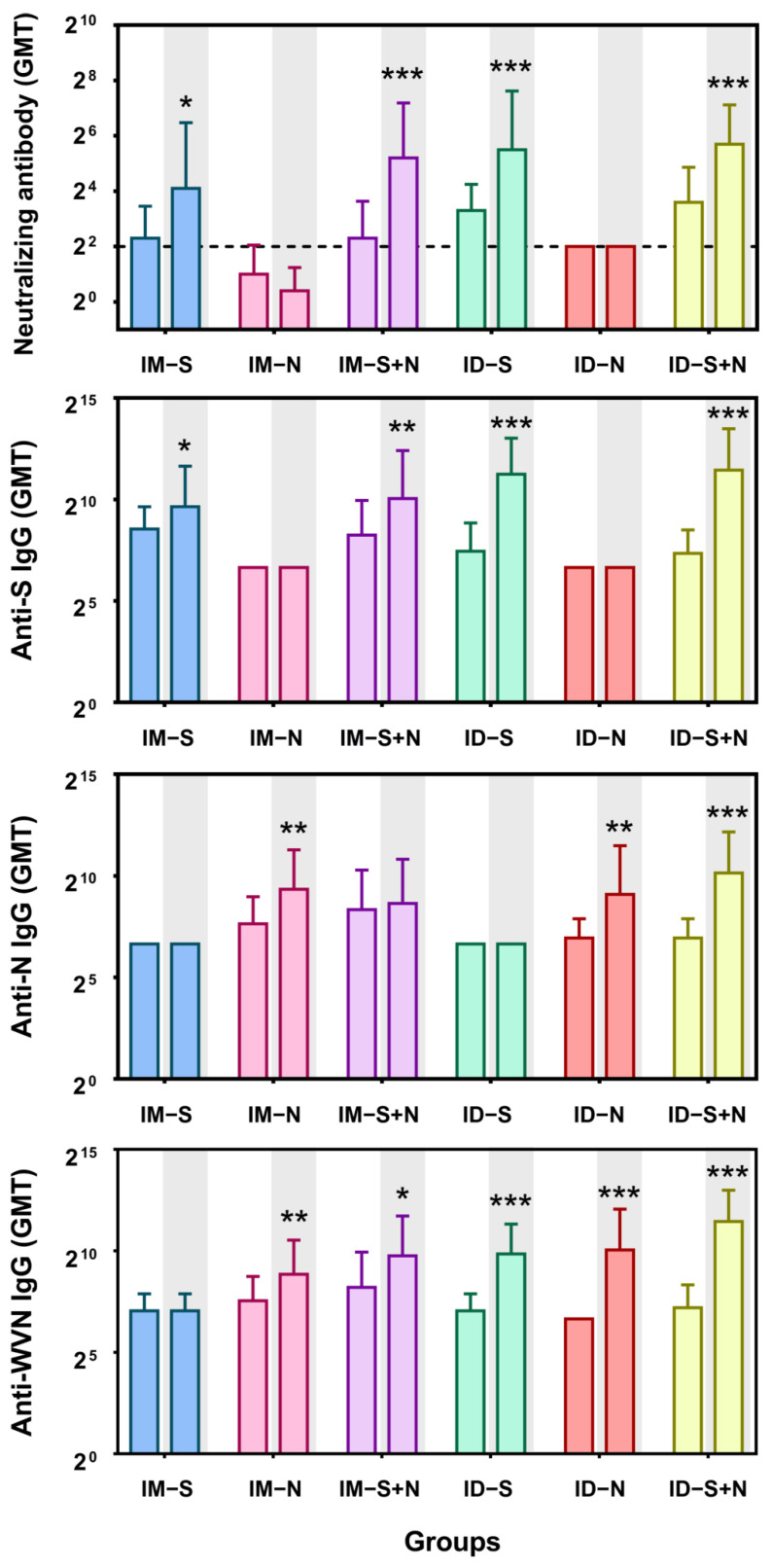
Levels of antibody production induced by experimental recombinant proteins after ID and IM immunization. The titers of neutralizing antibodies and IgG antibodies against S, N, and whole virus antigens (WVN) were detected 14 days after the primary and booster immunizations. The white background is 14 days after the first immunization (primary), and the gray background is 14 days after the second immunization (booster). IM-S: immunized with the S protein via the IM route; IM-N: immunized with the N protein via the IM route; IM-S + N: immunized with the S/N mixed protein via the IM route; ID-S: immunized with the S protein via the ID route; ID-N: immunized with the N protein via the ID route; ID-S + N: immunized with the S/N mixed protein via the ID route. The dotted line in the neutralizing antibody graph represents the positive threshold (neutralizing antibody titer of 4). The differences between the immunized groups and the control group were analyzed by *t*-test. *, *p* < 0.05; **, *p* < 0.01; ***, *p* < 0.001 compared with the control group (*n* = 10 per group).

**Figure 3 vaccines-10-01996-f003:**
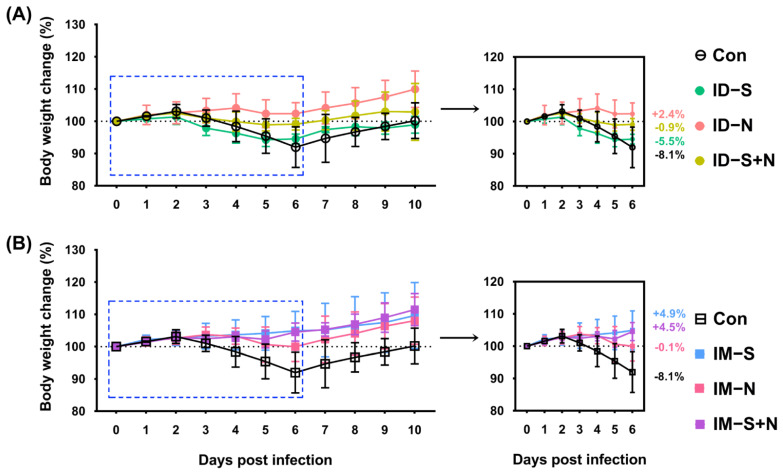
The changes in weight of hamsters immunized with the recombinant proteins via the ID and IM routes during SARS-CoV-2 challenge. (**A**) Challenge after ID and (**B**) challenge after IM. Changes in body weight after virus challenge within 10 days (left) and 6 days (right) are shown. Con: control group; ID-S: immunized with the S protein via ID administration; ID-N: immunized with the N protein via ID administration; ID-S + N: immunized with S/N mixed protein via ID administration; IM-S: immunized with the S protein via IM administration; IM-N: immunized with the N protein via IM administration; IM-S + N: immunized with S/N mixed protein via IM administration. The average weight change of each group on the 6th day after challenge is indicated as a colored number. Data are means ± SDs (*n* = 10 per group).

**Figure 4 vaccines-10-01996-f004:**
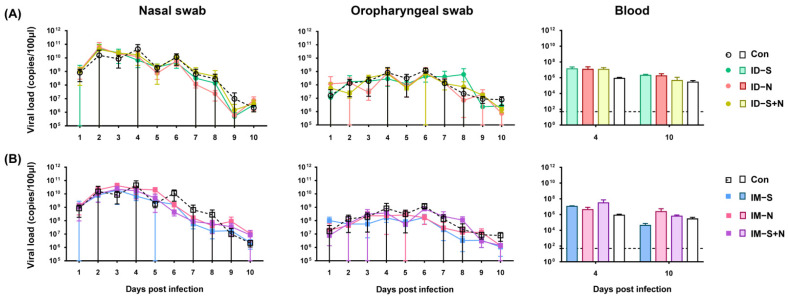
Viral shedding and viremia in hamsters immunized with the recombinant proteins via the ID and IM routes after SARS-CoV-2 challenge. (**A**) Challenge after ID and (**B**) challenge after IM. RNA was extracted, and the viral load was determined as copies per 100 μL. The dotted line in the figure indicates a PCR detection threshold of 50 copies/100 μL (*n* = 10 per group for nasal swabs and oropharyngeal swabs, and *n* = 3 per group for each blood-monitoring time point).

**Figure 5 vaccines-10-01996-f005:**
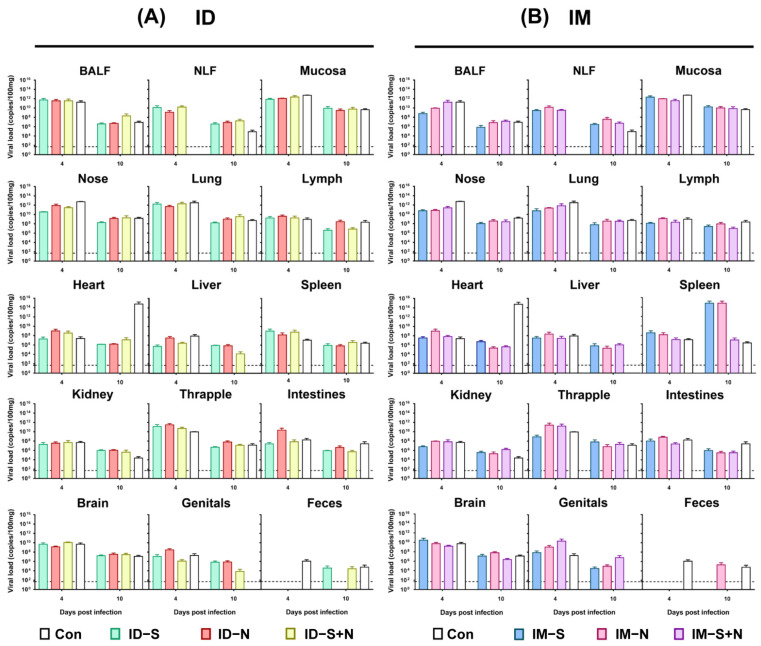
The viral loads in various tissues and organs of hamsters immunized with experimental recombinant proteins via the ID and IM routes after SARS-CoV-2 challenge. (**A**) Challenge after ID and (**B**) challenge after IM. RNA was extracted, and the viral load was determined as copies per 100 μL. BALF: bronchoalveolar lavage fluid; NLF: nasal lavage fluid. The dotted line in the figure indicates the detection threshold of the PCR method of 50 copies/100 μL (*n* = 3 per group for each time point).

**Figure 6 vaccines-10-01996-f006:**
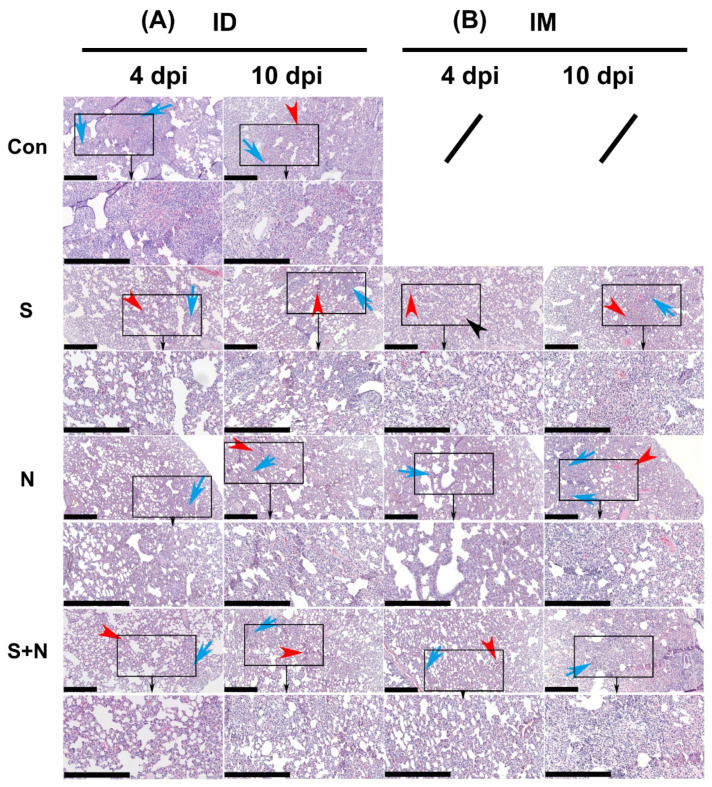
Pathological damage to the lungs in hamsters during viral challenge after immunization with the experimental vaccines via the ID and IM routes. (**A**) Challenge after ID and (**B**) challenge after IM. H&E-stained sections of lungs from virus-challenged hamsters. Tissue hyperemia and inflammatory cell infiltration detected at 4 and 10 days after challenge are indicated with red and blue arrows, respectively (scale bars are 0.4 mm). The organization in the black rectangle is magnified bigger and shown (towards black arrow) in below figure.

**Figure 7 vaccines-10-01996-f007:**
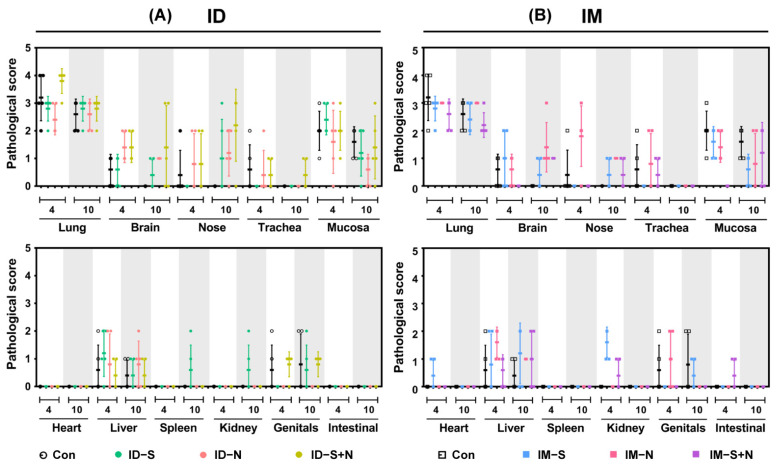
The pathological score of each tissue and organ from hamsters during viral challenge after immunization with the experimental vaccines via the ID and IM routes. (**A**) Challenge after ID and (**B**) challenge after IM. Scoring criteria: -, no abnormality (0); ±, slight infiltration of inflammatory cells (1); +, mild damage with inflammatory cell infiltration (2); ++, massive tissue damage with inflammatory cell infiltration and local vascular congestion (3); +++, serious tissue necrosis with inflammatory cell infiltrates and hyperemia (4); ++++, on the verge of death or dead (5).

**Figure 8 vaccines-10-01996-f008:**
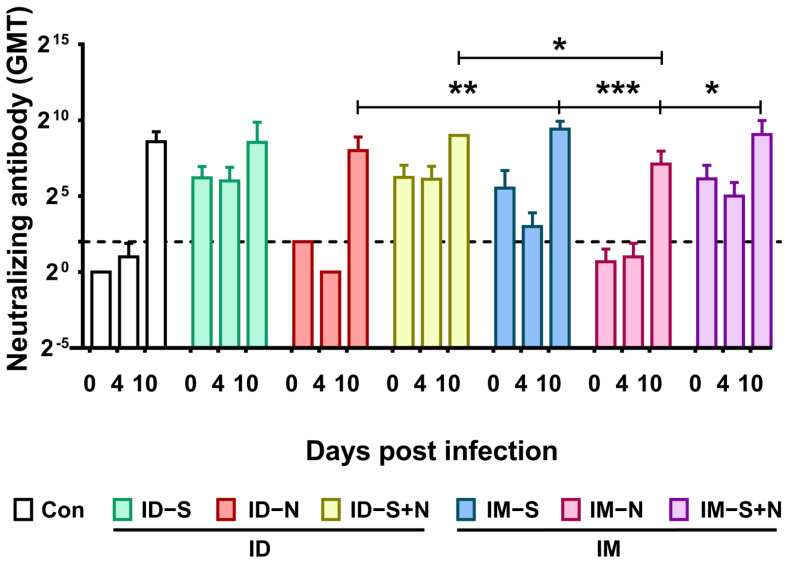
Neutralizing antibody levels in hamsters during viral challenge after immunization with recombinant proteins via the ID and IM routes. The titers of neutralizing antibodies were detected at 0 (14 days after the booster), 4, and 10 days after viral challenge. The dotted line in the neutralizing antibody graph represents the positive threshold (neutralizing antibody titer of 4). The differences between the immunized groups and the control group were analyzed by two-way ANOVA. *, *p* < 0.05; **, *p* < 0.01; ***, *p* < 0.001 (*n* = 10 per group (14 days after the booster or 0 days after viral challenge) and *n* = 3 (4 and 10 days after viral challenge) for each time point).

## Data Availability

Not applicable.

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
