# Peer review of "Evaluation of Immunogenicity and Clinical Protection of SARS-CoV-2 S1 and N Antigens in Syrian Golden Hamster"

_vaccines, 2022, doi:10.3390/vaccines10121996_

Round 1

Reviewer 1 Report

Analysis of the Syrian golden hamster as an evaluation model for COVID-19 peptide vaccine immune effectiveness and clinical protection

This is an excellent attempt by the author to evaluate a novel concept of a combined antigen vaccine against COVID-19. The present study is designed meticulously and well executed. The article is well-written and provides cutting-edge information and an add-on to the vaccine effort against SARS-CoV-2 infection. Following are the specific comments to further strengthen the manuscript.

Specific comments,

1.     Line 31 in the abstract, Interestingly, the neutralizing antibody level increased significantly after viral challenge. What does this statement for? Please rewrite and make it in context, it is hard to understand for readers, is it for a combined S and N peptide vaccine?

2.     In the introduction, line 39, please provide the latest data instead of January 2022. Now that the person infected has nearly doubled to the data provided, please refer WHO dashboard for the latest epidemiological data.

3.     Please consider attaching information on figure legends along with the figure legend heading, otherwise, these information looks hidden in the text.

4.     In the figures, adding figure sub number will be helpful for the better understanding of the readers. For example, Figure 3. (A) Challenge after ID and (B) Challenge after IM.

5.     Whether this vaccine will be considered for any further studies as not shows many advantages over protection and pathological damage.

6.     Is there any comparative study of the present peptide vaccine with the mRNA vaccine?

Reviewer 2 Report

I want to congratulate the authors for what was quite a lot of work. But the end result of their labours is a paper that I find confusing because it is effectively looking at two different issues at the same time - (1) the possible suitability of Syrian hamsters as a lab model for evaluating potential SARS -Co 2  vaccines and :  (2) new polypeptide vaccines.

The end result was that neither worked very well. The test vaccines did not appear to work, or only to induce antibodies but no protective effects, in the hamsters. After challenge there was little to distinguish between the responses of immunized hamsters and their unimmunized controls.

So I am left wondering a) how good are the hamsters for this work really and b) perhaps the polypeptides would work better in monkeys or even man.

I would have been interested first to see some study showing how one or more of the vaccines that we know work in man and monkeys can really work in hamsters. Then expand that to this study with the polypeptides whilst using a known successful vaccine as a positive control alongside PBS as the negative control.

I think there should have been much more discusion about the hamster model in the introduction; its strengths , weaknesses and unknowns.

Finally, why is the data about the global prevalence of Covid dating from January. This is October. Where has the paper been since then?

Reviewer 3 Report

The authors describe the Syrian golden hamster as an evaluation model for COVID-19 peptide vaccine based on the S and/or N proteins. The results suggest that the immune response induced by the experimental protein vaccines had no significant effect on the prevention of virus infection and pathogenicity, except for slight alleviation of weight loss. However, the animals do not show dramatic symptoms upon challenge (<10% of body weight loss) as was shown in this work and other papers describing Syrian hamster as a SARS-CoV-2 model. The title of the manuscript is somehow confusing, because the animal model is not the major subject of this work. In fact, the Syrian hamster as a SARS-CoV-2 model was described nicely by Imai et al. in 2020 (Syrian hamsters as a small animal model for SARS-CoV-2 infection and countermeasure development, published in PNAS), Sia et al. also in 2020 (Pathogenesis and transmission of SARS-CoV-2 in golden hamsters, published in Nature) and some others. The manuscript hardly mentions previous publications on the Syrian hamster as a model for SARS-CoV-2 infection and the use of this model for evaluation of the effectiveness of potential therapeutics. The authors should include previous publications on Syrian hamster, but also about other animal models, and their flaws.

Since the major subject of this manuscript is evaluation of potential peptide vaccine, the preparation of the protein/peptides used for immunization should be described in details. As it is, the manuscript has not sufficient value and significance.

The animal experiments were carefully planned with rational number of animals and design of immunizations, challenge, sample collection and animal euthanasia.

Other comments:

Introduction section: “…this study selected the S and N proteins, which have been reported to induce immune responses, and tested the two recombinant proteins alone or combined with Alum adjuvant to formulate experimental vaccines.” The paper does not describe combination of vaccine with and without Alum, only with Alum.

Line 76: please describe KMS-2 strain in more detail (date of collection, variant, is there an accession number from the GenBank...)

Line 81: please explain what are „Different types of recombinant proteins... “. How were the proteins prepared, in E.coli, CHO or something else. Which strain was used as a model/sequence for preparation of recombinant proteins? Was structural analysis done for recombinant proteins (multimers or monomers, glycosylation etc.)?

Line 96: What was the challenge dose?

Line 113: which SARS-CoV-2 virus strain was used for in neutralization assay?

Figures 2-6: text following the figure is probably meant to be a part of the figure legend, but it looks as part of the text.

Figure 5: if there are statistically significant differences, they should be indicated.

Figure 6: figures of lung sections should be larger to easier observe the difference.

In Discussion, the authors are comparing the effectiveness of the inactivated virus vaccine in the same model to the effectiveness of the protein/peptide vaccine. They emphasize the effect of other viral proteins present in the inactivated virus as the reason for better vaccine than single protein vaccine. However, this is only a part of the possible explanation. The inactivated virus contains other viral components that act as natural adjuvants. Also, the structure of the natural proteins often has a multimeric form which also enhance immune response. The authors should discuss this matter more.

There are several typos, please check the text carefully.

Round 2

Reviewer 2 Report

Thanks for your reply and comments to my first round of comments. I found them helpful and I think the manuscript is much better, and the title much more releva and accurate.

I have 3 more comments.:

1. On line 196 you write "A previous study reported significant weight loss among hamsters but no deaths after challenge". You must include this reference here - surely?

2. On line 268 you say "Consistent with the viral loads in the lung, pathological damage was 268 similar in all groups, including the control group (Figure 6)". But you do not tell us how you quantify this? If this is a subjective or qualitative sort of assessment then try to tell the reader how you did this.?

3. I think that your new text on the use of the hamster model is positive. However, I would have thought it more appropriate to place this in the introduction - to explain why you have gone for the hamster model. Its use as a model with other viruses is not necessary.

Reviewer 3 Report

The authors have acknowledged most of the comments.
